# The Influence of Surface Quality on Flow Length and Micro-Mechanical Properties of Polycarbonate

**DOI:** 10.3390/ma14205910

**Published:** 2021-10-09

**Authors:** Martin Ovsik, Michal Stanek, Adam Dockal, Petr Fluxa, Vlastimil Chalupa

**Affiliations:** Faculty of Technology, Tomas Bata University in Zlin, Vavreckova 275, 760 01 Zlín, Czech Republic; stanek@utb.cz (M.S.); a_dockal@utb.cz (A.D.); fluxa@utb.cz (P.F.); v_chalupa@utb.cz (V.C.)

**Keywords:** polycarbonate, flow length, surface quality, micro-mechanical properties, hardness

## Abstract

This study describes the influence of polymer flow length on mechanical properties of tested polymer, specifically polycarbonate. The flow length was examined in a spiral shaped mould. The mould cavity’s surface was machined by several methods, which led to differing roughness of the surface. The cavity was finished by milling, grinding and polishing. In order to thoroughly understand the influence of the mould surface quality on the flow length, varying processing parameters, specifically the pressure, were used. The polymer part was divided into several segments, in which the micro-mechanical properties, such as hardness and indentation modulus were measured. The results of this study provide interesting data concerning the flow length, which was up to 3% longer for rougher surfaces, but shorter in cavities with polished surface. These results are in disagreement with the commonly practiced theory, which states that better surface quality leads to greater flow length. Furthermore, evaluation of the micro-mechanical properties measured along the flow path demonstrated significant variance in researched properties, which increased by 35% (indentation hardness) and 86% by indentation modulus) in latter segments of the spiral in comparison with the gate.

## 1. Introduction

Within an injection mould, the melt goes through narrow channels, which support wall slip due to the mould being tempered to lower temperatures than the melt. Thus the polymer melt is cooled at the wall, which increases its resistance against flow. A different situation can be observed in the middle of the cross-section, where the temperature is still high and the surrounding polymer melt acts as an insulator. All these effects combined lead to fountain flow, which can be seen in Figure 1.

Fountain flow is a complicated event, which cannot be modelled in the same way as flow between two parallel plates. The velocity profile in absolute values resembles a bell, and the highest velocity of the flow is located in its centre. While the melt flows forward in the middle of the cavity, the melt at the walls flows against the main flow direction [1].

Many studies have been conducted on the topic of the influence of surface quality on the behaviour of polymers. However, only a handful of these publications were concerned with the effect of the surface quality, its replication ability and the applied processing parameters on the final product. Furthermore, no complex study analysing the influence of surface quality on the flow length of a polymer or the effect of flow length on the mechanical properties has yet been completed [1,2,3,4,5,6,7,8,9,10,11,12,13,14,15,16,17].

For example, the influence of mould’s surface on the flow length in thin-walled injection moulding was researched by Otsuka et al. [2].

Abdelkhalik et al. [3] considered the influence of various processing parameters on the qualitative characteristics of polymer parts manufactured by micro injection moulding (µIM). The processing parameters (injection speed, holding pressure, melt temperature and mould temperature) were tested on two polymers, i.e., polypropylene (PP) and acrylonitrile butadiene styrene (ABS).

The influence of melt temperature and injection pressure on the flow length was researched by Salimi et al. [4]. The goal of this publication was to examine the aforementioned behaviour in some of the most commonly used plastics, such as acrylonitrile butadiene styrene (ABS), polycarbonate (PC), polyamide 6.6 (PA 6,.6) and polyoxymethylene (POM). In this study, various channel depths were used.

Many scientists have examined and defined the polymer melt flow in an injection mould. Numerical studies of the polymer melt flow were discussed by Ershov et al. [5]. Moreover, a generalized 2D output model of the polymer melt flow was created by Pachner et al. [6]. In these studies, however, the influence of the surface roughness was not included. On the other hand, the influence of the wall roughness on a linear shear flow was investigated by Assoudi et al. [7]. Furthermore, Ebrahimi et al. [8] have studied the wall slip of molten polymers influenced by the surface roughness. Nevertheless, all of these studies were focused on extrusion, which does not require the use of high shear rate levels.

Injection moulding simulation models were discussed in studies by Zink et al. [9] and Hua et al. [10]. However, the surface roughness of the injection mould cavity was not taken into account. Recently, there has been growing interest in micro-injection moulding. In this technology, wall slip of linear polymer melts was observed by Gao et al. [11]; however, this knowledge cannot by directly applied to a common injection moulding process.

The wall slip during polymer flow can be partially caused by the cavity surface roughness. This property is also one of the key factors of the surface quality of the product. On account of this, the finishing operations are to machine the mould cavity, including runners. These technologies, such as grinding and polishing, are expensive and time consuming. There are comparative costs also with different manufacturing processes. As can be seen, machining costs steeply escalate with decreasing surface roughness values [12,13,14].

The surface roughness is generally expressed as Ra or average roughness and it is the arithmetic average value of the deviation of the trace above and below the centre line (Figure A1). The value of Ra is normally measured in micro-metres. ISO standards use the term CLA (Centre Line Average). Both are interpreted identically [8].

The goal of this study is the evaluation of micro-mechanical properties, which were measured along the flow of the polymer. For this endeavour, several chosen technological parameters and surface roughness of the spiral cavity were examined.

## 2. Materials and Methods

The preparation of the experiment dealt with planning of the manufacturing conditions of the specimens and the simulation of the injection moulding process for the chosen material (polycarbonate). The simulations were used to set the technological parameters of the injection moulding cycle.

Polycarbonate was chosen as the polymer for this experiment, and it was injected under three different injection pressures. At the same time, three plates with various surface roughness were used. In total, three groups of variables (pressure, surface quality and micro-mechanical properties) were chosen. Ten specimens were manufactured for each of these combinations.

### 2.1. Material

Polycarbonate commercially available as PC CALIBRE 302EP-22 manufactured by TRINSEO (Boehlen, Germany) was chosen as the tested material. This polymer is commonly used in practical applications and belongs to the group of amorphous polymers. The most important parameter affecting the polymer flow is the melt flow rate, which is 22 g/10 min for this polymer.

### 2.2. MouldFlow Analysis

Simulation of the injection moulding cycle for the chosen material was done in MoldFlow supplied by AUTODESK (San Rafael, CA, USA). Based on these simulations, some of the technological parameters of the injection moulding cycle were chosen.

### 2.3. Mould

Injection mould with simple construction enabling easy manipulation and replacement of test plates was used for the preparation of specimens. The dimensions of the test injection mould (200 × 200 × 12 mm) were selected with respect to the dimension of the cavity (spiral). This mould can produce one product at a time and provides the user with the ability to change the gate size. This mould was inserted into a universal frame, which can be used to prepare various test samples. Individual parts of the test mould can be seen in Figure 2a.

The cavity of the mould consists of a shaping plate and exchangeable test plate. The shaping plate forms the melt into a spiral, through which the flow length is observed. The test sample injected into the spiral can reach up to 2000 mm in length. The left side of the mould was tempered to required temperature. The same shaping plate was used for the entire duration of the experiment, while the exchangeable test plate located within the right side of the mould was switched for different specimens. Figure 2b displays the cavity section, in which it can be seen that the test plate exchange affected the flow length of the material by approximately 43%, which was the overall surface area of the test plate out of the entire cavity.

A sprue puller insert was designed to provide a simple way of changing the dimensions and the type of the gate. This insert guarantees not only the pull of the hardened melt remains, but also the use of four various gate types (1, 2, 4 and 6 mm). Rotation of the sprue puller insert enables the gate dimension change, and so the accurate position of this insert is given by locking sleeve. In order to reach the maximum possible flow length of the polymer, a film gate with 6 mm width was used. This width was set according to the dimensions of the injection moulded article, which can be seen in Figure 3.

During the experiment, three exchangeable test plates were used. Individual plates were labelled accordingly. The machining of the surface was done by milling, grinding and polishing. These operations and the subsequent quality of the surface can be seen in Table 1.

### 2.4. Sample Preparation

The specimens were manufactured by injection moulding machine ALLROUNDER 470 C GOLDEN EDITION supplied by ARBURG (Losburg, Germany).

The technological parameters of the injection moulding process were chosen according to the values gained from the injection moulding cycle simulations and the information provided by the material sheet of the tested polymer (this information can be found in Table A1). The values of holding pressure were set according to the injection pressure. On the one hand, injection pressures lower than 200 bar were not used, since it resulted in non-measurable flow length. On the other hand, pressures higher than 800 bar resulted in mould opening and overflow between the individual threads of the spiral.

The tested material was dried in accordance with parameters found in material sheet. The required conditions were given by the granulate supplier (Table A2). The drying process was conducted on THERMOLIFT 100-2 manufactured by ARBURG (Losburg, Germany). Granulate was delivered into the injection moulding machine by pneumatic suction.

### 2.5. Surface Quality

The measurements of the surface quality for individual test samples were done by contactless optical device Talysurf CLI 500 with software supplied by Taylor Hobson (Leicester, UK). The parameters that were used for the measurements can be found in Table A3.

Each spiral was divided into three parts (beginning, middle and the end). These areas can be seen in Figure 2c.

### 2.6. Micro-Indentation Properties

Micro-mechanical properties were measured on Microindentation Tester MHT_3_ manufactured by Anton Paar (Graz, Austria). The measurements were done according to EN 14557 standard. The settings of the measurements can be found in Table A4. Up to five points along the flow path were examined; starting from the gate and finishing at a given length (0 mm, 26 mm, 53 mm, 66 mm and 70 mm). Each point was measured ten times, and the values were further used to calculate the arithmetic mean and the standard deviation. The main micro-mechanical properties of interest were indentation hardness and indentation modulus.

Indentation hardness (*H_IT_*, Figure A2) was calculated as the maximum load (*F_max_*) on the projected area of the hardness impression (*A_p_*) [18,19].
(1)$$H_{IT}=\frac{F_{max}}{A_{p}}$$
(2)$$A_{p}=23.96\cdot h_{c}^{2}$$

The indentation modulus (*E_IT_*) was calculated from the plane strain modulus (*E^*^*) using an estimated Poisson’s ratio (*ν_s_*) of the sample (Polymer 0.3 to 0.4) [20,21].
(3)$$E_{IT}=E^{*}\cdot (1-v_{2}^{s})$$
(4)$$E^{*}=\frac{1}{\frac{1}{E_{r}}-\frac{1-v_{i}^{2}}{E_{i}}}$$
(5)$$E_{r}=\frac{\sqrt{\pi }}{2\cdot C\sqrt{A_{p}}}$$
where *E_i_* is the elastic modulus of the indenter (diamond 1141 GPa), *E_r_* is the reduced modulus of the indentation contact, and *ν_i_* is the Poisson’s ratio of the indenter (0.07).

## 3. Results

### 3.1. Injection Moulding Simulation

Figure 3 displays the comparison of simulation results and the injected specimen. The test sample was prepared from PC, which was injected under 600 bars of pressure with the use of polished test plate. The surface roughness is not considered in simulations, thus the surface was pre-set to values that reflect the reality of polished surface. The comparison clearly shows that the flow length was approximately the same. Both the simulation and the injection moulding process adopted a gate with 6 mm width, which was the same as the width of the spiral.

### 3.2. Flow Length

It is a generally accepted fact in industrial practice that better surface quality of the injection mould provides improved flow of the polymer melt within the mould cavity. For this reason, the finishing operations of the cavity, for example polishing, are done. This increases the final cost of the injection mould. This publication shows that it is not always required to polish the cavity of the mould in order to gain improved polymer melt flow.

The evaluation of the flow length dependence on the injection pressure and the surface quality of the cavity can be seen in Figure 4. The test plates completing the spiral shaped cavity were finished by three technologies (milling, grinding and polishing) and as such had different surface roughness (Ra 1.6, Ra 0.8 and Ra 0.1). As is evident from the results, the flow length was affected by both the injection pressure and the surface roughness.

The longest flow length was observed in cavities with surface roughness Ra 1.6, except for specimens injected with pressure 200 bar. On the other hand, the shortest flow length was measured in specimens injected in cavities with surface roughness 0.1. While the injection into cavity with surface roughness 0.1 with 800 bar pressure led to 70.2 mm flow length, the flow length was improved by 2% for the cavity with roughness 1.6, i.e., 71.2 mm. When other injection pressures were used, the difference in flow length was similar.

These results were confirmed in measurements of numerous other materials that were tested. The assumption arising from these measurements is that rougher surfaces, i.e., milling, are created by structure with more irregularities that are not fully filled by the flowing polymer. This phenomena can be seen in Figure A3. Thus, these depressions remain filled with air (an insulant) which creates a thermal barrier between the injection mould (mould temperature 80 °C) and the polymer (melt temperature of the injected polycarbonate is 260 °C).

The injected polymer slips along this air barrier, which results in slower cooling and thus greater flow length than in moulds with improved surface quality, i.e., polishing. These results were confirmed by the surface profile measurements, which also showed improved surface quality in the already produced part, in comparison with the mould in which it was produced (Figure 4).

### 3.3. Micro-Mechanical Properties

The evaluation of the polycarbonate’s micro-mechanical properties was done through the measurement of its melt flow. Specimens prepared with 800 bar injection pressure were measured in five points unevenly placed along the flow direction. The first point was placed at the gate (0 mm), while the other points were placed at 26 mm, 53 mm, 66 mm and 70 mm. The measurements were done ten times at each point.

The P-h indentation curves measured at a given distance from the gate (0 mm, 53 mm and 70 mm) for the specimens manufactured in moulds with varying surface roughness (Ra 0.1, Ra 0.8 and Ra 1.6) can be found in Figure 5. The curves, which are used in calculations of the mechanical properties, demonstrate the elastic–plastic behaviour of the materials.

The main parameter characterizing the properties of the surface is the indentation hardness. As can be seen in Figure 6, the indentation hardness differs in each predetermined point for every type of surface. The mould with polished surface showed the highest indentation hardness at the gate (182 MPa) and lowest indentation hardness at 66 mm, which was 135 MPa. This decrease was 35%. At the end of the specimen the hardness was measured to be 165 MPa. The overall decrease between the beginning and the end of the specimen was 10%.

Concerning the grinded plate (Ra 0.8), the indentation hardness was measured to be 161 MPa at the point placed at gate (0 mm). Subsequently, the hardness decreased at 26 mm to the value of 133 MPa, which meant a 21% decrease. After this point, the hardness began to improve up to 158 MPa measured at the end of the specimen (70 mm).

The milled plate (Ra 1.6) proved to have similar tendencies to the grinded plate. The indentation hardness was 155 MPa at the gate (0 mm) and later at 26 mm decreased by 14% to 136 MPa. Following measurement points displayed a gradual increase of hardness, which resulted in 143 MPa at the end of the specimen (70 mm). The difference between the gate (0 mm) and the end of the specimen (70 mm) was 8%.

Similar tendencies as with injection pressure 800 bar were measured with injection pressure 400 bar (Figure 7). In this case, the flow length was 53 mm. Polished cavity (Ra 0.1) displayed hardness of 180 MPa at the gate. The hardness then fell to 146 MPa at 26 mm, which was a decrease of 23%, and subsequently increased to 165 MPa at the end of the specimen. The difference in hardness between the gate and the end of the specimen was 10%.

Results of the measurements of the grinded cavity (Ra 0.8) also showed similar tendencies to the measurements of the polished cavity. At the gate (0 mm), the hardness was measured to be 158 MPa, while the point at 26 mm displayed a slight decrease of hardness (151 MPa). The hardness subsequently increased at the end of the specimen (53 mm) to 172 MPa. The increase of hardness between the gate and the end of the specimen was 9%.

The hardness measured at the milled cavity (Ra 1.6) varied only slightly in individual measurement points.

Besides the hardness, the indentation modulus is also an important parameter, which can be used to characterize the micro-mechanical properties of a surface. The results of the measurements of the indentation modulus for samples injected with 800 bar pressure can be seen in Figure 8. It is evident that the results of indentation modulus correspond with the results of the indentation hardness.

The measurements of the polished mould (Ra 0.1) show that the highest modulus (3.45 GPa) was found at the gate (0 mm). Further along the flow direction, the modulus was decreasing up until the point of measurements at 53 mm, at which the value was 0.71 GPa. The variance between the indentation modulus at the gate and the one at 53 mm was 86%. The indentation modulus measured at the end of the specimen (70 mm) was 3.23 GPa.

The cavity with grinded (Ra 0.8) and milled (Ra 1.6) surfaces were measured to have similar tendencies throughout the results of the indentation modulus. The highest indentation modulus value was measured at 66 mm.

The results gained by the measurements of the specimens prepared with 400 bar injection pressure can be seen in Figure 9. Test samples moulded in cavities with Ra 0.1 and Ra 0.8 displayed similar tendencies in the measurements of indentation modulus. At the gate (0 mm), the indentation modulus was measured to be 3.40 GPa (Ra 0.1) and 3.16 GPa (Ra 0.8), while the results gained at the middle of the specimen were 1.88 GPa (Ra 0.1) and 2.50 GPa (Ra 0.8). The decrease of the indentation modulus was 81% for surface with roughness Ra 0.1 and 26% for surface with roughness Ra 0.8. The indentation modulus at the end of the specimen (53 mm) was 2.71 GPa and 3.28 GPa. The shaping plate with Ra 1.6 displayed only slightly varied results in the individual measuring points.

### 3.4. Surface Quality

Examples of surface quality measurements of both the mould and the surface of the specimen can be seen in Figure 10. The aforementioned examples of measurements are demonstrated on the injection mould with surface roughness Ra 0.8, which was used to manufacture the test sample with 800 bar injection pressure. Figure 10a displays the scan of the mould’s surface quality, while Figure 10b shows the scan of the specimen’s surface. The surface of the injection mould was finished to surface roughness Ra 0.82 µm and Rz 5.26 µm, nevertheless the values of the surface which replicated to the specimen were Ra 0.58 µm and Rz 4.37 µm. Figure 10 contains the 3D scan demonstrating the differences in surface quality. Furthermore, the profile of the injection mould’s surface can be seen in Figure 10e, while the profile of the specimen’s surface is shown in Figure 10f. The comparison of these profiles shows the variance in surface quality, as well as the lack of the replication concerning the biggest surface irregularities.

The influence of mould’s surface quality on the product’s surface quality is an important parameter which is necessary in understanding of the polymer behaviour within the cavity. The dependence of flow length on surface quality can be seen in Figure 11 and Figure 12; the former represents injection moulding with 400 bar pressure, while the latter was done with 800 bar. As is obvious from the results, the imprint of mould’s surface on the final plastic product depends on various factors, e.g., distance from the gate, roughness of the cavity and injection pressure. The measurements of the surface quality along the flow direction showed that the values differed considerably; the surface roughness was lower at the beginning of the sample than it was at the end. The difference in surface roughness measured at the gate and at the end of the specimen was significant. The test sample prepared in cavity with surface roughness Ra 0.1 was tested to have up to six times rougher surface than the cavity. On the other hand, the specimens injected within the cavity with testing plates finished to Ra 0.8 and Ra 1.6 proved to have lower roughness than the testing plates themselves. The most precise replication ability was found in specimens injected in cavity containing testing plate with Ra 0.8; and so the boundary between the negative and positive replication was surface roughness Ra 0.8. In practical terms, there is no reason to finish the surface of the mould to Ra 0.1, since Ra 0.4 or Ra 0.8 is sufficient. The application of this research could lead to significant savings impacting both the cost of the mould and the final product. The surface roughness measurements correspond with the results of flow length and mechanical properties.

## 4. Conclusions

This study describes the effect of surface quality of the mould on the flow length of tested polymer (polycarbonate), as well as the influence of flow length and processing conditions of the injection moulding on the mechanical properties, which were tested along the entire flow direction. The injection mould test plates complementing the spiral shaped cavity were finished with three different technologies, which led to differing surface roughness of the mould (milling—Ra 1.6, grinding Ra 0.8 and polishing Ra 0.1).

The results of the flow length measurements are at odds with general knowledge of this subject, which states that better surface quality leads to better melt flow. On the contrary, the results of this study indicate, that worse surface quality leads to improved flow length. The difference in flow length, between the mould finished with polishing (70.2 mm) and milling (71.2 mm), was approximately 2%. The results of measured properties were quite similar for each tested injection pressure.

The measurements of the surface layer show significant differences in mechanical properties at varying distances from the gate to the end of the specimen. The highest values of indentation hardness were measured at the gate, while the lowest values were measured further in the specimen. The indentation hardness decreased by 35% approximately in the middle of the specimens. Similar tendencies were measure for each tested injection pressure.

The measurements of the replication ability of the injection mould on the final plastic part confirm the previous results of the flow length and the mechanical properties. The replication of the mould’s surface on the final product is affected by the flow distance from the gate as well as the surface quality of the injection mould. The distance from the gate proved to have a significant influence on the final surface quality, i.e., the further it was measured from the gate, the worse surface quality was measured. The specimens manufactured in cavity with surface roughness Ra 0.1 displayed worse surface roughness, while the test samples produced in cavity with surface roughness Ra 1.6 showed improved surface quality. The boundary between the positive and the negative replication of the surface quality was found in cavity with surface roughness Ra 0.8. These results have a significant effect in the industrial practice. The flow length and the surface quality replication results show that it is not necessary to finish the cavity of a mould to surface roughness Ra 0.1, since the finish to Ra 0.4 or 0.8 is sufficient. This will contribute to considerable savings in the cost of a mould.

This publication has significant implications for the manufacturing of injection moulded parts in industry. The finishing operations of the cavity are not needed to improve the flow length, unless the final aesthetics of a part are important. These findings could lead to significant cost savings, which can make up to 100% of the final mould cost. The measurements of the mechanical properties indicate that the most important factors of mould design are the placement of gate and the flow path of the polymer. The gate should be placed in such points of the mould where the flow path is short, which results in less varied hardness at further distance from the gate.

The submitted results will be used as a stepping stone for further research with different types of materials (amorphous/semi-crystalline). This endeavour should demonstrate and verify the influence of the injection mould surface and the injection moulding parameters on the flow length. A significant result for the practical application is without a doubt the influence of flow length on the final properties of the tested material. The differing flow length manifested in varying mechanical properties at measuring points set along the flow direction.

## Figures and Tables

**Figure 1 materials-14-05910-f001:**
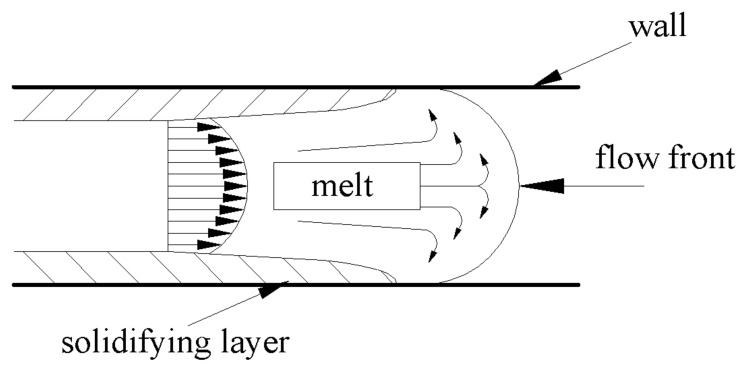
Fountain flow in the injection mould.

**Figure 2 materials-14-05910-f002:**
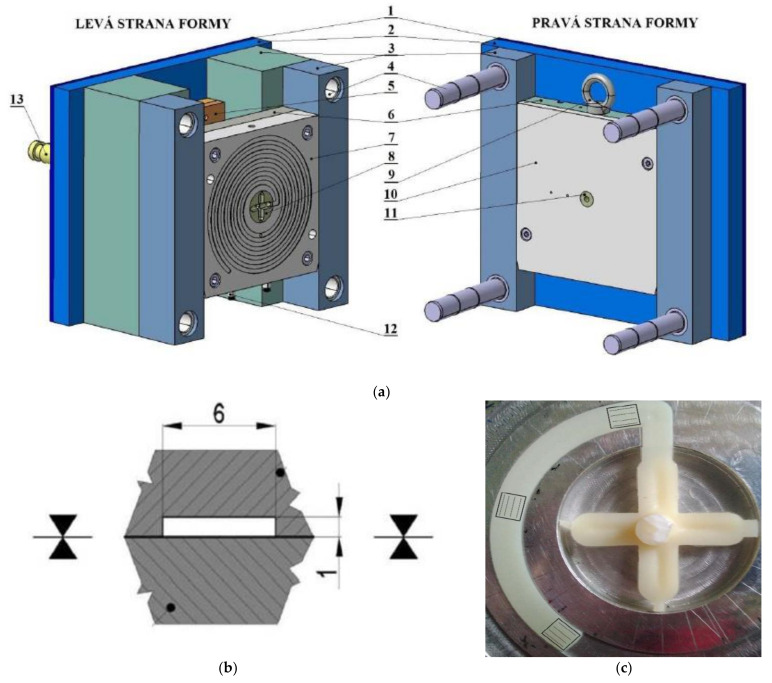
(**a**) 3D model of the test injection mould; (**b**) cross-section of the cavity; (**c**) area of surface quality measurement: 1—heat insulation board 2—clamping plates, 3—spacer block, 4—guiding elements, 5—ejection system, 6—backing plate, 7—cavity plate, 8—Sprue puller insert, 9—lifting eye bolt, 10—test plate, 11—sprue, 12—hose nipple, 13—ejector bar.

**Figure 3 materials-14-05910-f003:**
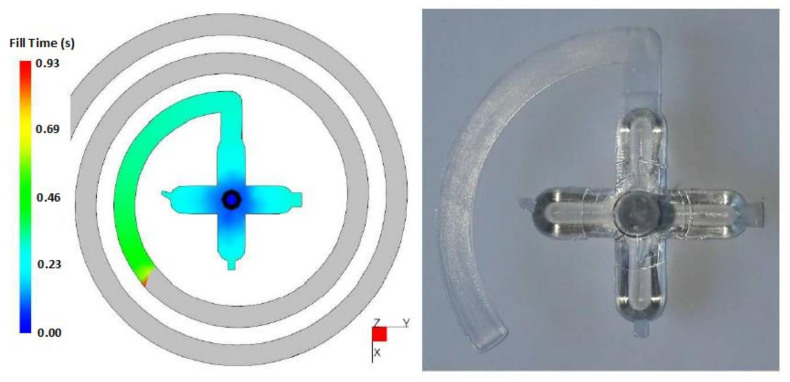
Comparison of the flow length displayed in simulation and the physical test sample (material PC, injection pressure 600 bar).

**Figure 4 materials-14-05910-f004:**
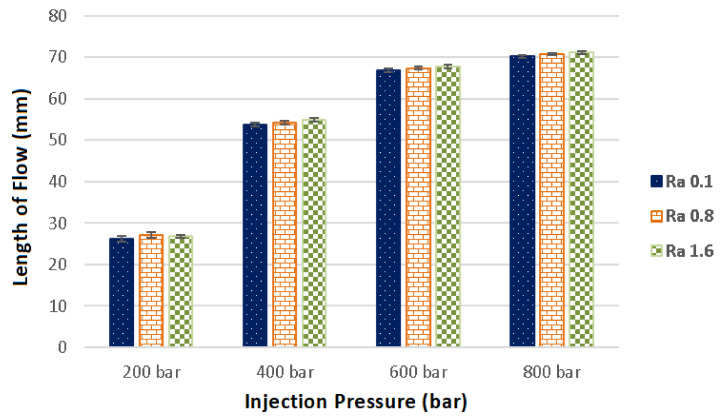
Dependence of flow length on the injection pressure for each surface roughness (Ra 0.1, Ra 0.8, Ra 1.6).

**Figure 5 materials-14-05910-f005:**
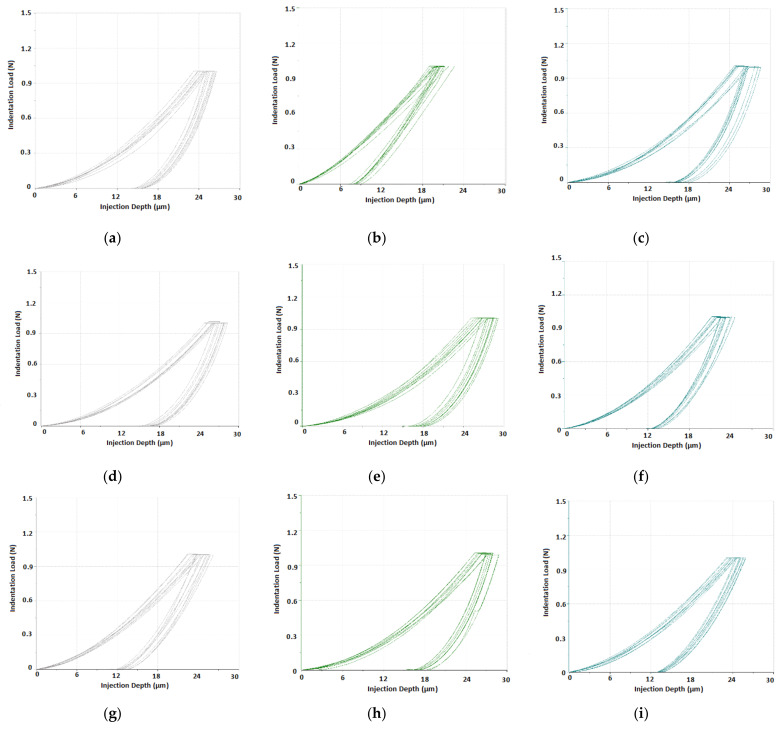
The indentation P-h curves at surface quality with different flow length: (**a**) Ra 0.1, 0 mm; (**b**) Ra 0.1, 53 mm; (**c**) Ra 0.1, 70 mm; (**d**) Ra 0.8, 0 mm; (**e**) Ra 0.8, 53 mm; (**f**) Ra 0.8, 70 mm; (**g**) Ra 1.6, 0 mm; (**h**) Ra 1.6, 53 mm; (**i**) Ra 1.6, 70 mm.

**Figure 6 materials-14-05910-f006:**
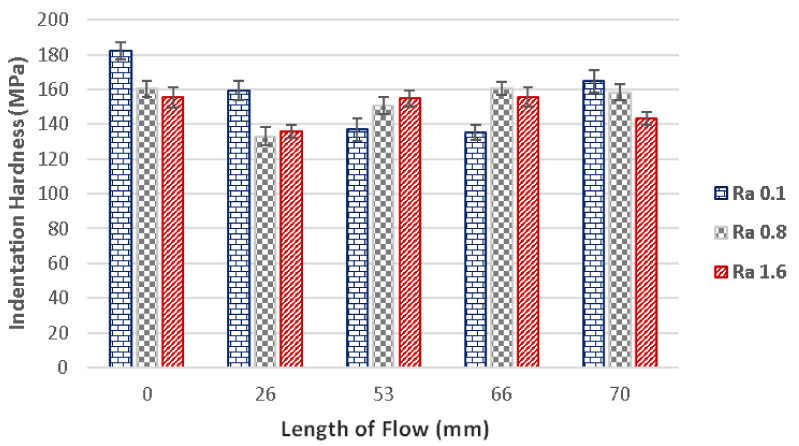
The dependence of the flow length on the indentation hardness for surfaces with roughness Ra 0.1, Ra 0.8 and Ra 1.6 and injection pressure 800 bar.

**Figure 7 materials-14-05910-f007:**
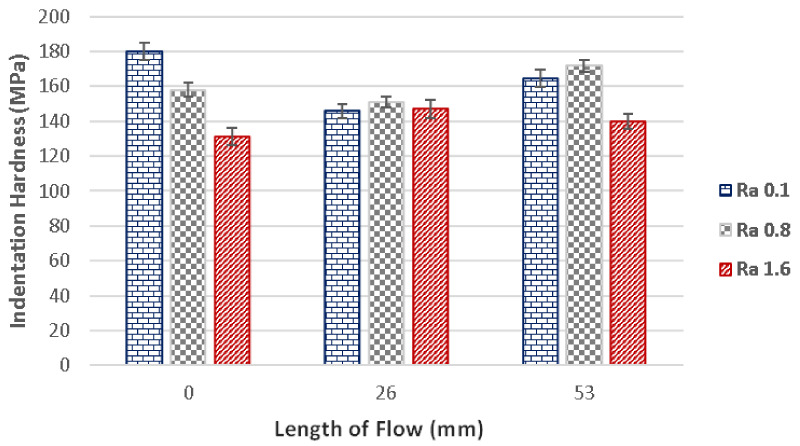
The dependence of flow length on the indentation hardness for following surface qualities: Ra 0.1, Ra 0.8, Ra 1.6 and injection pressure 400 bar.

**Figure 8 materials-14-05910-f008:**
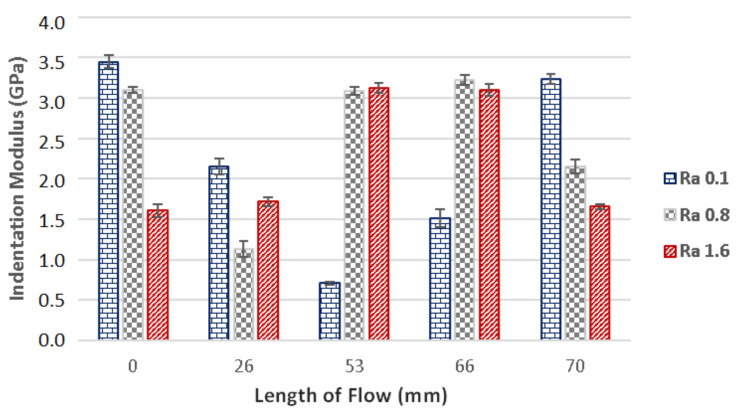
The dependence of the flow length on the indentation modulus in cavities with following surface quality: Ra 0.1, Ra 0.8, Ra 1.6 and 800 bar injection pressure.

**Figure 9 materials-14-05910-f009:**
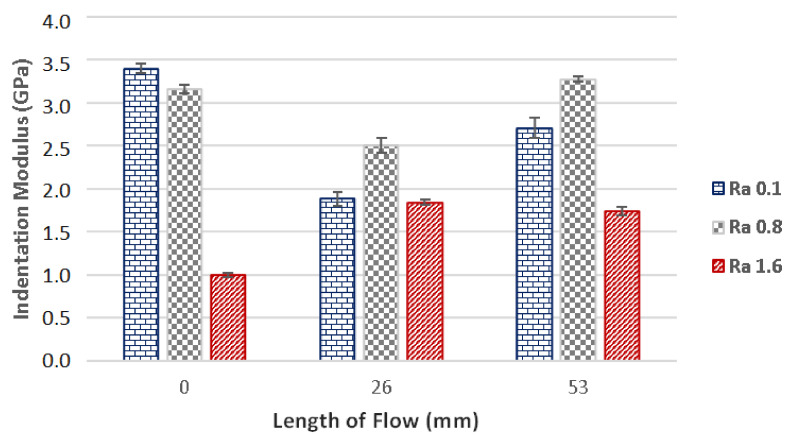
The dependence of the flow length on the indentation modulus for plates with following surface roughness: Ra 0.1, Ra 0.8 and Ra 1.6. The injection pressure was 400 bar.

**Figure 10 materials-14-05910-f010:**
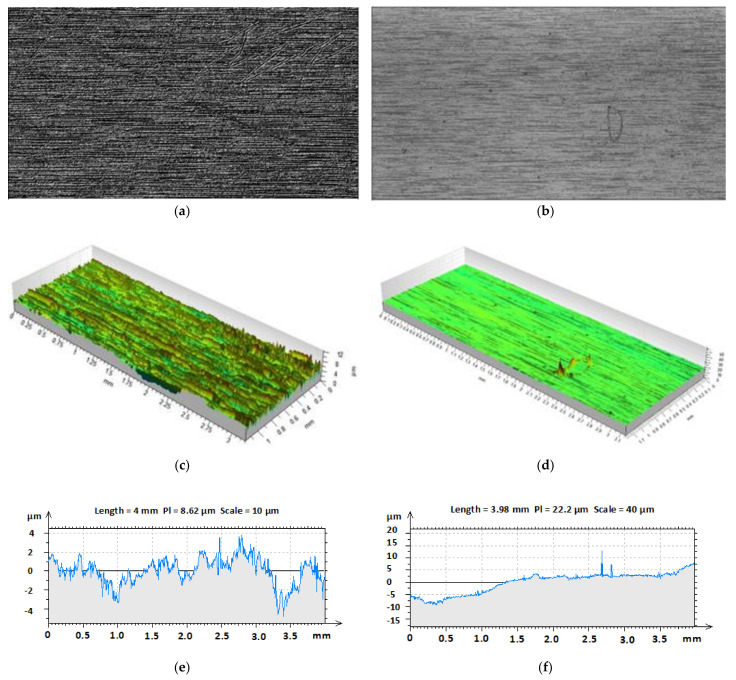
The surface quality—injection pressure 800 bar, flow length 0 mm: (**a**) mould; (**b**) product; (**c**) mould; (**d**) product; (**e**) profile of the mould; (**f**) profile of the product.

**Figure 11 materials-14-05910-f011:**
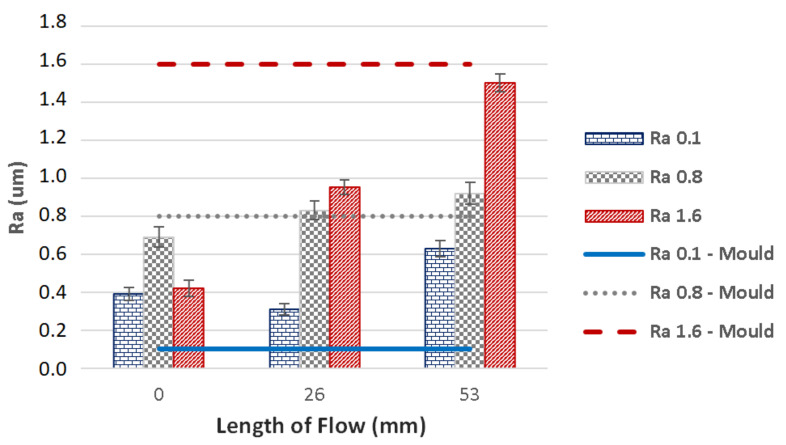
Dependence of the flow length on the surface quality of the mould and the product—injection pressure 400 bar.

**Figure 12 materials-14-05910-f012:**
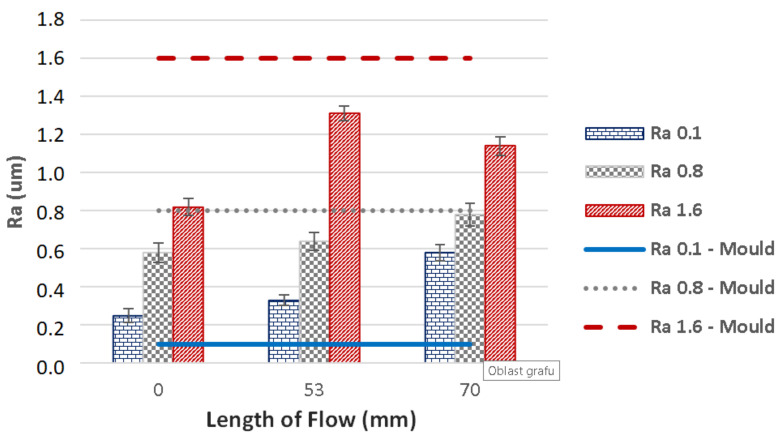
Dependence of the flow length on the surface quality of the mould and the product—injection pressure 800 bar.

**Table 1 materials-14-05910-t001:** Parameters of the surface of the injection mould.

Test Plates Label	Type of Machining	Surface Quality
Plate 1.6	Milling	Ra = 1.6 µm
Plate 0.8	Grinding	Ra = 0.8 µm
Plate 0.1	Polishing	Ra = 0.1 µm

## Data Availability

The data presented in this study are available on request from the corresponding author.

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
