# Peer review of "The Influence of Surface Quality on Flow Length and Micro-Mechanical Properties of Polycarbonate"

_materials, 2021, doi:10.3390/ma14205910_

Round 1

Reviewer 1 Report

This manuscript has provided a lot of data with figures and tables, however, it is poorly organized and lack of important control experiments.  The authors should focus on the most important part of their studies and put many tables and figures into Supporting/Supplementary Information.  I suggest reconsider it after major revision.  The details of the issues are listed below.

  1. In Abstract, last line, “…, which were 35% higher (respectively 86%) in latter segments of the spiral than at the gate.” I don’t understand what this means.
  2. It is necessary to explain the definition of surface quality Ra before Figure 2.
  3. Figure 3, Figure 4, Figure 5, and Figure 6 can be combined as one figure.
  4. Table 2, Table 3, Table 4, Table 5, Figure 7, all of them should be placed into Supporting Information.
  5. In Figure 8, this structure has different geometries, some very short straight channels and a long spiral channel, why did the authors choose to study this relatively complex geometry? Did the authors test their hypothesis on simple straight channel and/or simple spiral channel? Please start from the simple geometry, at least use it as control, then extend to more complex geometry.
  6. Figure 10, Figure 11, and Figure 12 can be combined as one figure.
  7. Figure 13 and Figure 14 can be combined as one figure.
  8. It is impossible to read the numbers on the x- and y-axis in Figure 15, Figure 16 and Figure 17, and there is no unit label on any axis. These three figures should be combined as one figure, and please remake all of the axes and add units to them.
  9. It seems like the key figures in this manuscript are Figure 18 to Figure 22, and the conclusion is on page 16, so the authors used 12 pages before that part to show other information, that is not the way to write a research paper. Please rewrite the manuscript to emphasize what is really important in this study.

Author Response

Thank you very much for your comments regarding our paper. They are quite useful for us and help us improve the quality of the prepared paper. We gratefully accept your comments and the changes according to your recommendation were included in the corrected version of the paper. I hope the correction will meet your requirements and you will accept it. Concerning English language, the language proof of the text was done by native speaker.

Reviewer 2 Report

The manuscript titled “The Influence of Surface Quality on Flow Length and Micro-Mechanical Properties of Polycarbonate” is an interesting piece of work by Ovsik and coworkers. They described the influence of polymer flow length on mechanical properties of polycarbonate by measuring various micro-mechanical properties including hardness and indentation modulus. This is an interesting piece of work and will surely benefit industrial and academic chemists. Therefore, I support publication of this work in Materials. However, prior to that, I suggest authors to consider following points:

  1. It is interesting that the flow length was longer by 3 % at maximum for rougher surfaces, but shorter in cavities with polished surface. What could be the possible reason for this observation?
  2. Page 2, line 37-42: Cite suitable references.
  3. Reference numbering given in the text are not matching with list given at the end of MS. For example, Otsuka et al. is cited as [2] which is wrong. It should be [13]. Same issue I found with Salimi et el. [4], Ershov et al. [5], Pachner et al. [6], Assoudi et al. [7] etc. Authors are suggested to throughly check this issue in the manuscript.
  4. References should be reformatted according to the journal style.
  5. Figure 8 (left), 15, 16 & 17: Scales are not clear, please insert a high resolution image.
  6. Authors might consider transferring some of the Figures and Tables to the supplementary information.

Author Response

(The authors gave the same response as above.)

Reviewer 3 Report

The manuscript is interesting however it is faraway from the focus of materials journal. It is more technical work than to be accepted in the journal of Matreials. The manuscript can not be accepted in its present form due to the following major comments:

  1. The focus of this work is not going well with what is required by Materials journal
  2.  The manuscript is more technical work than scientific one.
  3. The discussions of results should be subborted with indeep scientifc explanations. Speculative discussion is not helpful.

Author Response

(The authors gave the same response as above.)

Round 2

Reviewer 1 Report

This revision is much better, still need careful proof-reading since so many changes are made.  It can be published after language and format check.

Reviewer 3 Report

The manuscript has been modified to be much better to be accepted in its present form. The authours did most of required and major revisions.  

The manuscript can be accepted in its present form.